# Development of an artificial intelligence-enhanced warfarin interaction checker platform

**Monther Abdolmohsin Alsultan**[1]*, **Mohammed Alabdulmuhsin**[2], **Deema AlBunyan**[3]

**1** Department of Pharmacy Practice, King Faisal University, College of Clinical Pharmacy, Al-Ahsa, Kingdom of Saudi Arabia, **2** Department of Computer Sciences, King Faisal University, College of Computer Science, Al-Ahsa, Kingdom of Saudi Arabia, **3** Department of Management Information Systems, King Faisal University, College of Computer Science, Al-Ahsa, Kingdom of Saudi Arabia

* malsultan@kfu.edu.sa

## Abstract

Warfarin is a common anticoagulant drug for thrombo-prophylaxis in stroke and venous thromboembolism, which has many advantages but also some disadvantages including narrow therapeutic window, vast drug interactions (and wide variability with foods/herbs), as well as unpredictability of pharmacodynamics and/or kinetics. Complicating factors can present as challenges for the outpatient clinicians trying to strike that balance due to the potential consequences of over or under dose anticoagulation with associated increased risk of bleeding and/or thromboembolic events, respectively. Because warfarin interactions can drastically affect therapeutic outcomes, patient to healthcare provider communication regarding such potential drug-drug or diet-warfarin interactions is crucial for compliance with the medication and achieving successful treatment. Furthermore, language barriers cause low patient satisfaction scores and poor quality/safety health care. In fact, the advancement and improvements in healthcare technology promise artificial intelligence (AI) as one of ideal options to optimize delivery of health care. The goal of this study is to develop Warfa-Check, a bilingual AI-based web app that matches both speakers of Arabic and English. The application helps users recognize potential warfarin-associated drug interactions with a simple user interface that accepts text, picture or voice commands. Warfa-Check, developed with Python and Flask as well as OpenAI's GPT-4 API with natural language processing tools trained to correctly interpret outbound warfarin interactions. Multiple validation methods and beta testing have been done to ensure that the app is data-driven, as well color coded alerts for interaction severity provide clear feedback to end-users. This easy-to-use application helps patients identify drug interactions in both English and Arabic. Warfa-Check represents a valuable avenue for improving the safety of our residents, simplifying medication management in high-risk individuals and streamlining workflow. Future development plans are to develop into other anticoagulants, and integrate with Electronic Health Records (EHRs).

**Data availability statement:** The code and all programming scripts can be accessed through the following link: https://github.com/Xor01/warfa-check.

**Funding:** This work was supported through the Ambitious Researcher Track by the Deanship of Scientific Research, Vice Presidency for Graduate Studies and Scientific Research, King Faisal University, Al Ahsa, Saudi Arabia (Grant No.KFU250390). The funders had no role in study design, data collection and analysis, decision to publish, or preparation of the manuscript.

**Competing interests:** The author has declared that no competing interests exist.

## Author summary

Warfarin is a commonly prescribed blood thinner medication that can be very difficult to use for multiple reasons. Warfarin interacts with many other medications as well as food and herbs, some of which are a common part of the diet in specific regions (like grapefruit) leading to unwanted effects like bleeding or stroke. In addition, language barriers in healthcare delivery can lead to adverse events which need to be tackled for the effective management of warfarin use through excellent communication between health care providers and patients. In order to tackle these problems a bilingual artificial intelligence powered app has been developed "Warfa-Check," and this project called to design a user-friendly mobile application for patients that would be accessible in English, Arabic or both. The app provides text, image and voice input methods to check for interactions that simplify use whilst encouraging the safe management of this blood thinner. This represents the power of AI with an example in healthcare delivery, and specifically how it can enhance patient safety. Further enhancements are planned to broaden the app's reach into other classes of chronic medications, connection with Electronic Health Records (EHRs), as well as increased personalization.

## Introduction

Anticoagulants such as warfarin are frequently used to prevent and treat several blood clotting disorders. Despite the availability of newer anticoagulant drugs in the market, the usage of warfarin remains high. This is mainly due to the extensive efficacy data supporting its use, its affordability, and its widespread use among patients with atrial fibrillation. Though it offers several therapeutic advantages, it is also associated with certain limitations [1]. The narrow therapeutic index and significant interindividual variation in dose requirements are characteristics of warfarin [2]. Moreover, warfarin exhibits a limited range of effective dosage, numerous significant medication interactions, and an unpredictable safety record. Practitioners have effectively controlled its usage by employing the international normalized ratio (INR) readings and adjusting the dosage to minimize the likelihood of bleeding. Nevertheless, with the present therapeutic approach, patients, on average, maintain their target INR range for just 70% of the time, and incidents of bleeding do occur [1]. Achieving the most effective blood thinning with this medication is difficult in a clinical setting due to its numerous interactions with meals and other medications. Drug-drug interactions (DDIs) provide a substantial concern in drug prescriptions, varying in severity from benign to potentially leading to illness and death. A study was conducted to ascertain the frequency of possible drug-drug interactions (DDIs) in a public hospital in Saudi Arabia. 800 patient prescriptions were reviewed. The study indicated that 36.25% of the prescriptions examined exhibited at least one drug-drug interaction (DDI), with the majority of these interactions being of moderate intensity. The significant occurrence of DDIs in outpatient environments emphasizes the necessity for physicians and pharmacists to find potential solutions for identifying DDIs, as well as for patients to possess awareness regarding their medications subsequent to the dispensing of prescriptions by pharmacists [3]. Inadequate management of anticoagulation can subject patients to a heightened susceptibility to bleeding or thromboembolic problems, resulting from excessive or insufficient anticoagulation, respectively. Variations in the amount of vitamin K and proliferation of dietary supplements and herbal products can greatly impact warfarin's effectiveness [4].

There are also some considerations which may further limit patients' access to information about warfarin, thus adding to the challenges associated with its use in clinical practice. There

is often a lack of effective communication between physicians and patients regarding drug interactions. Ensuring patients receive accurate medication information is crucial for achieving the best possible outcomes in pharmacotherapy, medication compliance, and disease management. Patients who lack sufficient understanding of their medication or are given improper or incomprehensible information are less likely to comply with their treatment, resulting in suboptimal medication usage and reduced effectiveness of the therapy. Patients who possess a high level of knowledge are also more empowered and more inclined to engage in shared-decision making [5].

Language barriers present significant challenges in healthcare, affecting satisfaction levels among both healthcare providers and patients, compromising the quality of care, and potentially impacting patient safety. Such barriers lead to communication breakdowns, which decrease satisfaction for both parties and undermine healthcare quality and safety. Increasing evidence highlights the indirect impact of language barriers on healthcare quality. These barriers lower satisfaction for both patients and providers and hinder effective communication. Patients with language limitations are not only more likely to use additional healthcare services but also face a higher risk of adverse events. A recent study across six U.S. hospitals found that patients with limited English proficiency experienced adverse events at a higher rate than English-proficient patients [6]. Usually, information about medications may be found in English on the internet, and the majority of medical research papers are also published in English. Hence, in countries where English is not the primary language, a significant number of individuals encounter difficulties in accessing reliable data regarding their medications [7].

Healthcare technology is advancing at an impressive pace, accompanied by a variety of methods for assessing health technology [8]. Artificial intelligence (AI) is already widely used in healthcare. It is increasingly being integrated into healthcare delivery and holds great promise to improve diagnostic accuracy, treatment outcomes, and quality of patient care. Diagnosis of acute conditions; This seems to be one key area for using AI in healthcare. A classic example is to train AI algorithms for radiology application where it has been taught to recognize certain features in X-rays and CT scans as indications of disease or injury. Through this capability, healthcare practitioners can make more accurate diagnoses sooner and with correct detection that is increasingly non-invasive or targeted [9].

The objective of this study paper is to create a comprehensive application designed to help address several warfarin-related challenges encountered in clinical settings. This app can accommodate those who speak English or Arabic so that it simplifies the process of obtaining reliable and accurate medical information. With an aim at bridging the current gap in Arabic language resources available to people. The application aims to integrate sophisticated artificial intelligence (AI) technology to simplify the detection of warfarin common interactions with user-friendly features and improved healthcare delivery through minimizing medication error and overcoming language barriers.

## Results

This is the app link: https://warfa-check.onrender.com.

## App features and functionality

Fig 1 shows the main page of our online warfarin interactions checker, *Warfa-Check*, was developed based on a Python- based web application, which enables patients to check warfarin interactions with other medications, food and herbs in two languages: Arabic and English.

Below is a step-by-step process of how a patient uses the app (Figs 2 –7).

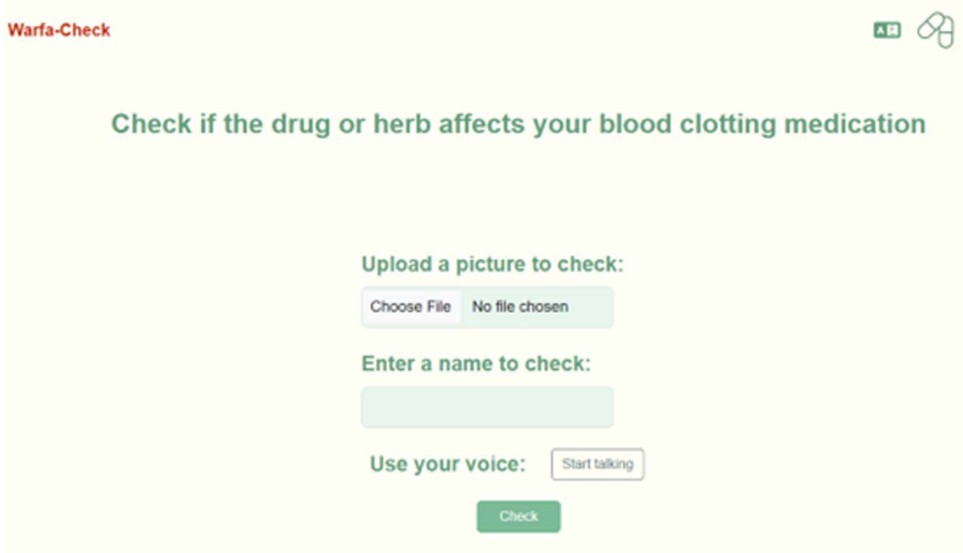

**Fig 1. The appearance of the online warfarin interaction checker page and the tools inside it.** The app's page contains the name of the app (Warfa-Check) and a title describing the app features (Check if the drug or herb affects your blood clotting medication). To facilitate patients' access to the information, patients can enter drugs, food or herbal names in three different ways: text, picture and voice.

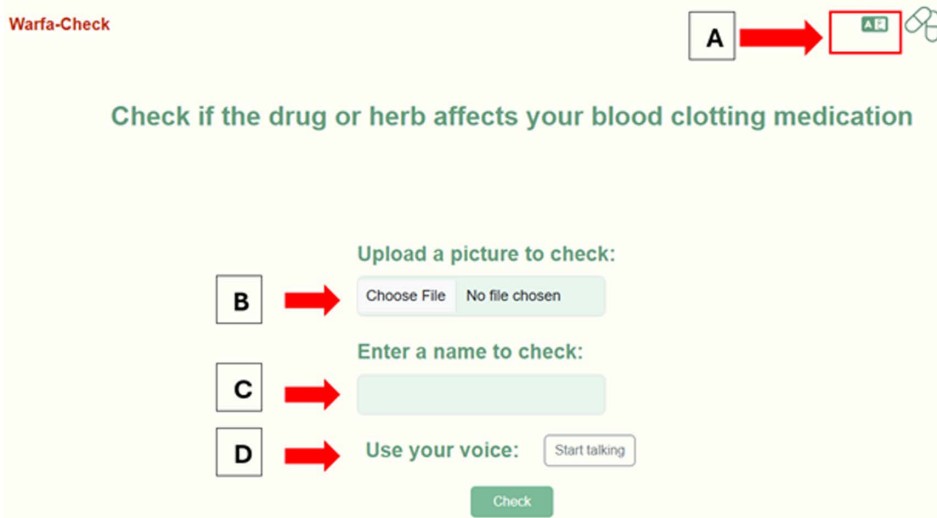

**Fig 2. Language options and methods of information entry.** In the top right corner, there is a button where users can choose from two languages: Arabic or English version **(A)**. Once the patient clicks on, it changes the page to Arabic or English based on the patient's preference. **(B)** Is an information entry where users either upload the picture of the medication, herb or food, or it can open the phone camera and take a photo of the medication. **(C)** Is a text entry where the users type in the medication, food or herbal name. **(D)** Is a voice entry where the users can speak out the medication, food or herbal name. After the users enter the medication, food, or herbal name, they can click on check to display the result.

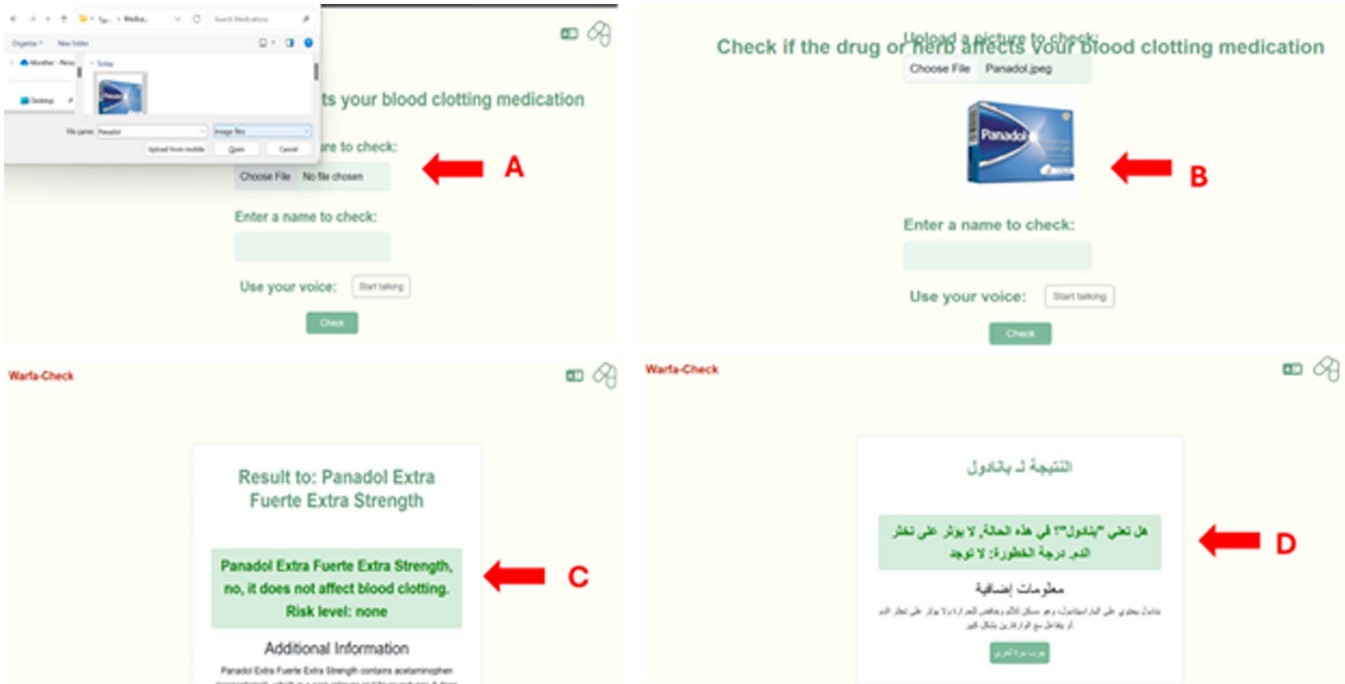

**Fig 3. Uploading pictures to the app and detection of the information in the picture.** The users can select "choose file" to upload the picture of any medication. The example here is "Panadol" as it is shown in A. In B, the app uploads the picture of the medication and then the app with the use of AI (chatGPT-4o), processes text, image, and voice inputs to recognize drug names and retrieve relevant interaction with warfarin or not (**C**). Additionally, the app gives a brief description of the selected medication to inform the user about this medication. Since the app has the Arabic version, it displays the information in Arabic as well (**D**).

## App validation and user satisfaction scores

Below is a summary of the survey results (user satisfaction scores):

- **Number of respondents:** 37 (7 patients, 12 pharmacists, 5 healthcare professionals, and 13 in other roles).

- **Overall satisfaction score:** 76% rated the application as highly satisfactory (scores of 4 or 5 on a 5-point Likert scale).

- **Website performance:** 70% of respondents agreed that the website performed smoothly without technical issues.

- **Usability and satisfaction:** 75% found the application easy to use and navigate.

- **Feature effectiveness:** 73% felt that the application's features (such as drug interaction detection and bilingual support) were effective and valuable.

- **Visual and information clarity:** 87% rated the visual design and information presentation as clear and engaging.

- **Overall impact:** 87% of respondents agreed that the application would positively impact patient safety and pharmacist workflows.

  Some of the respondents comments include the following:

- Suggestions for simplifying the drug-food interaction for convenient user reading.

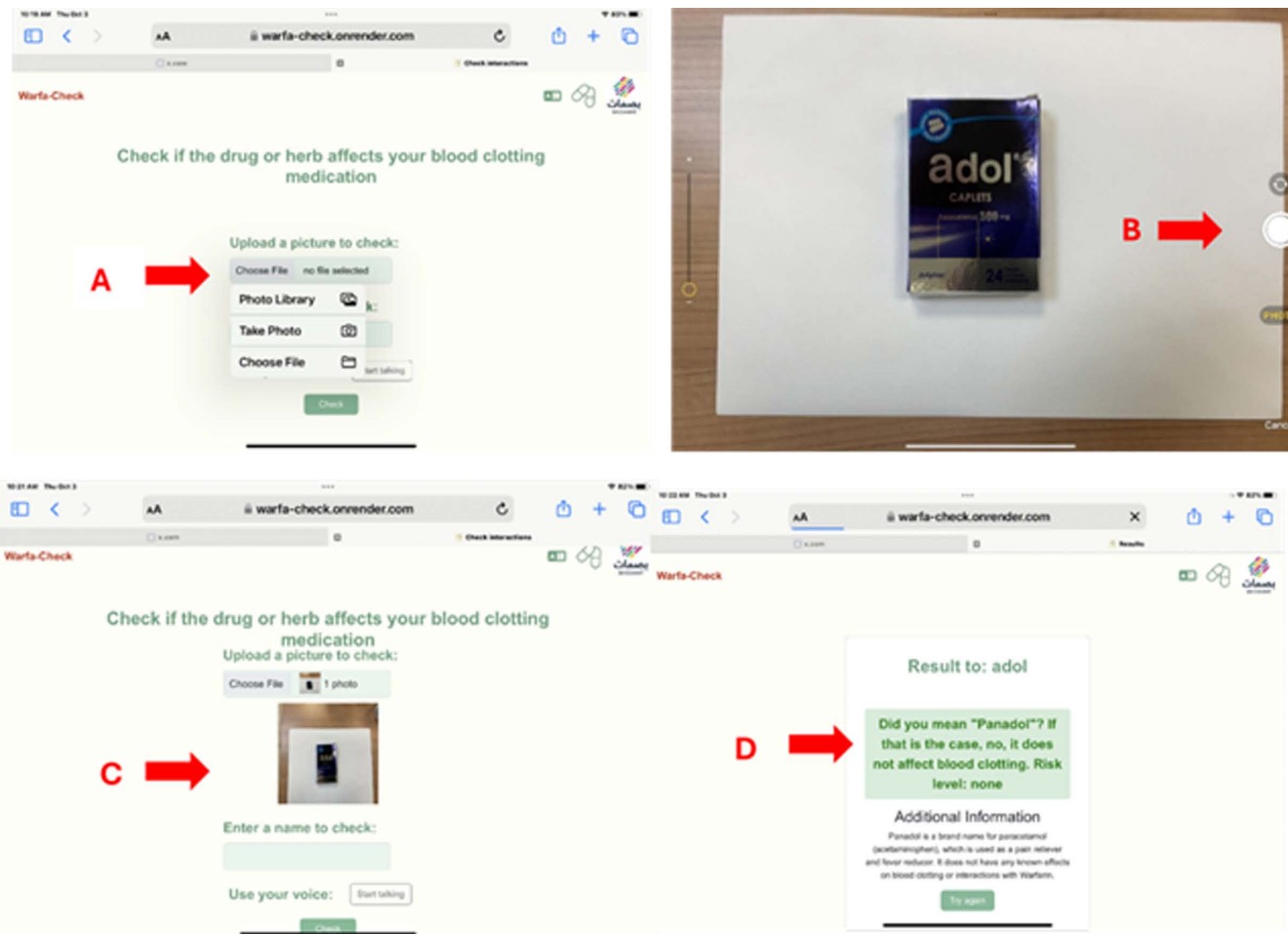

**Fig 4. Detecting the information of the taken picture and displaying the interaction information: the users can select "choose file to upload a picture of any medications as it is seen in (A and B).** Then the app uploads the picture **(C)**. The app with the help of Ai, will recognize the name of the medication in the picture and then give recommendations about its interaction with warfarin and this is the same when the user selects Arabic languages as more details is explained in Fig 3.

- Suggestions to add references to interaction information for better transparency.

- Positive feedback highlighting the accuracy and effectiveness of the platform.

## Discussion

The Warfa-Check app describes a major step towards giving patients and pharmacists the power to take control of warfarin therapy. The app then showcases a user-friendly interface, personalized for the Arabic speakers, with thorough visible and audial information on possible warfarin interactions can be revealed through text or picture formats as well as voice commands. Given the narrow therapeutic index and far-flung interactions with food, herbs, and other medication of the commonly used warfarin this app may have a significant influence on patient safety especially in high-risk patients for complications [10]. By providing patients with this essential information in real-time, patients may better make treatment plan decisions and improve adherence to treatment regimens, ultimately resulting in potentially reduced adverse events such as bleeding or thrombosis [1].

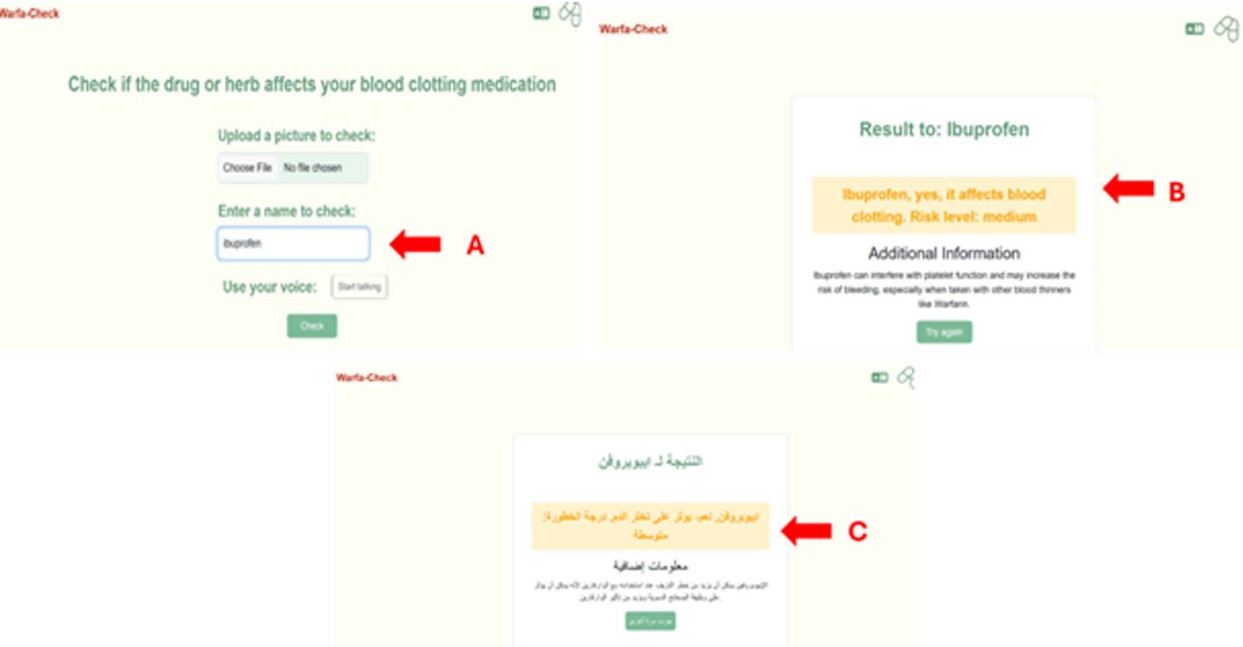

**Fig 5. Entry of the information manually and displaying the results.** Another option the users can use in this app is typing the medication manually (**A**), example here "Ibuprofen". The AI assistance in app works to recognize this medication from the database and then gives the user the interaction type with important information about this drug interaction with the warfarin (**B**). The user can enter the Arabic name of the medications as it is shown in the picture and then the app recognizes the medications to display the information in Arabic (**C**).

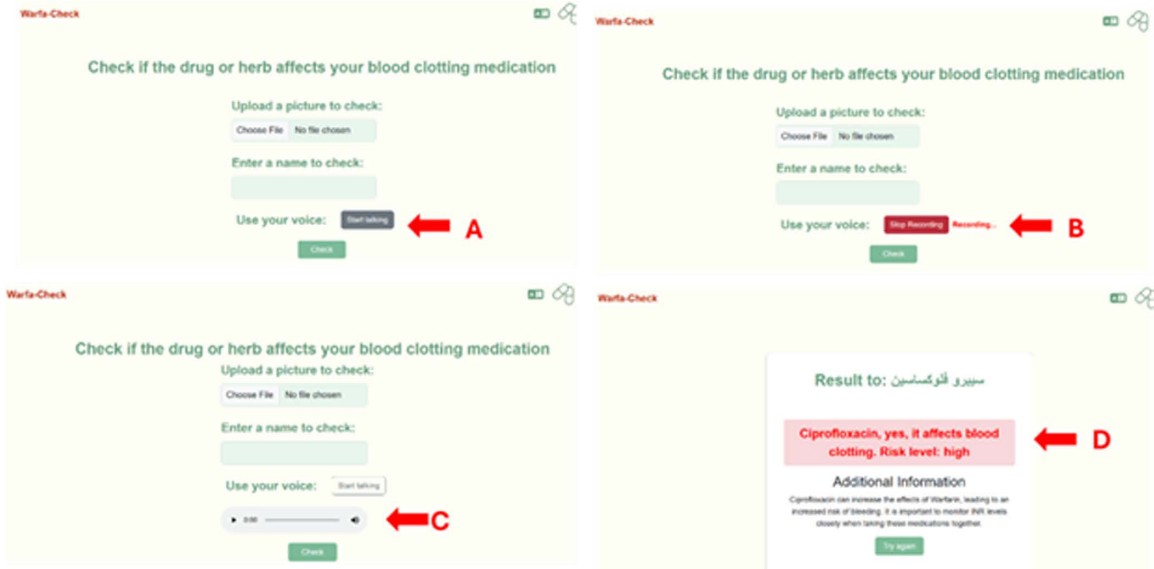

**Fig 6. Entry of information via user's voice and detection of the entered information.** In this app the users can enter the medication name using the voice feature (**A**). By clicking on "Start speaking", the user can speak out the name of the medication (**B**). As an example here, the ciprofloxacin name was spoken out. Once the user stops the recording, the user can listen to their record to check if it is correct or not (**C**), and by clicking on the check button, the app recognizes the medication name with assistance of AI to display the information needed for the users (**D**). This feature also supports the Arabic version, when the user speak in Arabic the app recognizes the drug name and display the information in Arabic.

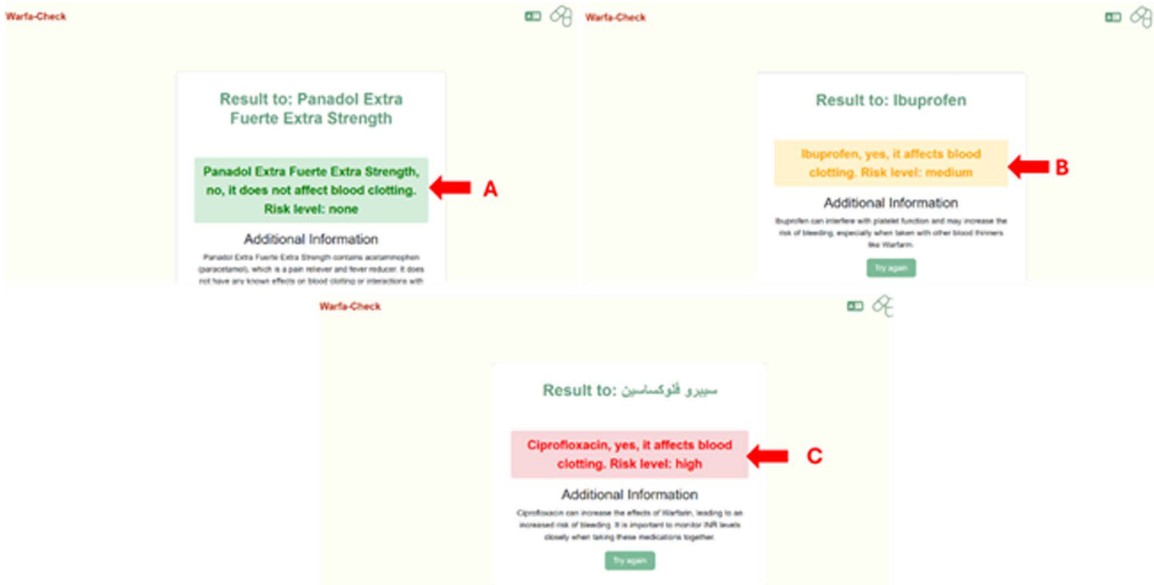

**Fig 7. Color coding the displayed information for easily understanding of the information.** The display information is color coded based on the severity of the interaction between warfarin and other medications, food or herbs. Green indicates no interaction or the risk is none **(A)**. Orange indicates medium interaction **(B)**. Red indicates high or major interaction. The color coding of the displayed information will keep patients' confusion as minimal as possible **(C)**.

This app is regarded as an important asset for pharmacists to simplify the process of recognizing drug interactions with food and drugs. These alerts often overwhelm pharmacists who may have to manually cross reference potential interactions in a few different databases or rely on their experience at the time, leading to missed dangerous combinations of drugs (because people mess things up). The idea is that Warfa-Check uses AI to support and enable the steps of this process, so pharmacists can concentrate on more clinical work tasks. The use of artificial intelligence in this application allows to detect interactions much more accurately and reliably, as early warnings may result in worse results for the patient [4]. Furthermore, the integration of a dual-language feature helps eliminate one of the largest communication challenges faced in English [7].

The wider implications of the use of AI in this way show the possibilities of artificial intelligence serving as aids to healthcare professionals. By automating the work of this process, such as checking interactions together, AI can do most of the grunt work and reduce the amount of manual labor a pharmacist has to do to prevent mistakes [3]. In addition, free tools like Python and ChatGPT not only provide pharmacists with an affordable way to create custom apps that meet individual or patient needs but also offer a creative route for people to explore other helpful technologies. They democratize app development and empower pharmacists with very little programming experience to take part in the care of the patient by creating personalized digital health tools [5,11].

**Future directions.** Although Warfa-Check is already a comprehensive tool for detecting warfarin interactions, several ways in which it could be improved are considered as next steps to maximize its interactivity. This would increase the relevance of the app to a wider patient population by extending it to interactions with other anticoagulants and common chronic medications. In addition, personalized interaction assessments could be conducted as well by integrating patient-specific information including current medication use, dietary practices and warfarin-metabolizing genetic factors such as CYP2C9 and VKORC1 variants [4]. This is an avenue for adding push reminding patients of upcoming doses or the known risks of their interactions — and to enforce the adherence a step further [2]. In addition, you might

use machine learning algorithms to continue to learn from the user's input and patterns of interaction that increase accuracy of this mobile app in the long run. Engaging with health professionals on expanding and validating the app's interaction database could help it remain a valuable resource within an ever-changing pharmacological world. 3) Enabling integration with EHR systems would allow the app to share interaction data with healthcare providers, for further improving patient care and safety [5,12].

## Materials and methods

### App development process

#### Design phase.

- User Interface (UI) Design: The UI design to be simple and accessible, ensuring that patients with different background technical skills could use the apps. The UI was designed to be clean, with an easy switch between languages; Arabic and English (Arabic and English), and simple input methods for checking warfarin interactions with other medications, food and herbs.

- User Experience (UX) Considerations: UX design was patient-centric focused to enable prompt information gathering efforts.. TThe app offers multiple input modalities (text, image and voice) to accommodate the different preferences or constraints for each user. Results are color-coded to illustrate clearly the severity of interactions.

#### Development tools.

- Programming Language: Python was used as the language for backend development and programming of the app. Python comes with a wide range of libraries which assists in AI integration, web application development.

- Frameworks and Libraries: 1. Flask: A micro web framework was utilized to handle the following: web routing, form submissions, and API calls. 2. OpenAI (GPT-4o API): with integrated natural language processing capabilities to identify drug, food and herbs names and analyze interactions (providing detailed information about the interactions and recommendations) using AI.

- HTML/CSS and Bootstrap: HTML/CSS was used for designing the frontend, ensuring responsive design that works consistently across devices. Bootstrap was used for styling and layout, giving a user-friendly interface.

#### Integration of AI.

- AI Algorithms: The app uses natural language processing (NLP) models powered by OpenAI's GPT-4o API fine-tuned on a collected and cleansed warfarin interaction dataset to discover potential interactions between warfarin in combination with other substances (medications, food, herbs.) ChatGPT-4o takes text, image, and voice as inputs to recognize drug, food herbs and names and then retrieve relevant interaction information and their properties associated with it. Fine-Tuning chatGPT-4o model in this is to allow and provide new data that is possibly never recognized before such as Arabic drug names and what are their scientific names.

### Database construction

Data sources.  Data from reliable drug databases have been collected to ensure the app provides accurate information. To ensure that the AI model can recognize medication names commonly interacting with warfarin, data was scraped, including images of required

medicines, from pharmacy websites. Afterward, data was cleaned to remove any irrelevant or unwanted information.

To finetune the AI model several steps were necessary. We passed these images through the ChatGPT-4o vision model and extracted their medicine or drug names, which then created to JSON Lines file with text that has been identified along with their respective scientific name. This file was consumed during the fine-tuning stage in order to allow for precision AI model identification, as well as response back of such drugs related queries.

**Data validation and maintenance.** The data were rigorously checked for accuracy of the information and cross reference the information with trusted sources such as lexicomp and uptodate. Additionally, to ensure the accuracy of the system in identifying drug interactions, beta testing was conducted with healthcare professionals and non-health care professionals. This involved real-world testing scenarios to assess interaction detection accuracy, usability, and reliability. Continuous feedback was incorporated iteratively to refine the platform.

## Testing and validation

**Beta testing.** The app was tested by healthcare professionals and non health care professionals to identify any issues or areas for improvement. To evaluate the user satisfaction of the platform, we conducted an anonymous survey with health care professional and non health care professionals.. Participants were invited to test the application and complete feedback from assessing various aspects if the application. The survey included questions covering five domains:

1. Website performance – Assessing speed, reliability, and responsiveness.

2. Usability and satisfaction – Evaluating ease of navigation, user-friendliness, and overall satisfaction.

3. Feature effectiveness – Assessing the usefulness of the drug interaction checker and bilingual functionality.

4. Visual and information clarity – Evaluating the layout, design, and clarity of information presented.

5. Overall impact – Assessing the perceived value and impact of the application on healthcare practices.

Participants rated each item on a 5-point Likert scale ranging from 1 (strongly disagree) to 5 (strongly agree). Additionally, responders were ask to leave comments to provide qualitative feedback.

**Security measures for handling medical information.** The platform was created mainly to demonstrate how AI can detect warfarin interactions with drugs, food and herbs, simply. The platform carries no personal identification information, or sensitive records. All user interactions are completely anonymous and there is no data retention, as queries are processed instantaneously. Because the tool works by not storing or sharing data, there are no specific security measures required at this time for protecting medical information.

**Feedback incorporation and iterative improvements.** The feedback taken was used for testing to refine the app, making improvements through multiple iterations.

## Acknowledgments

The author acknowledge the Deanship of Scientific Research, King Faisal University, Al-Ahsa, Saudi Arabia, for the support. Additionally, the authors thank King Faisal University, College of Computer Science and Nation Center for Giftedness and Creativity Research at King Faisal University for their resources and support.

## Author contributions

**Conceptualization:** Monther Alsultan.

**Data curation:** Mohammed Alabdulmuhsin, Deema AlBunyan.

**Methodology:** Mohammed Alabdulmuhsin, Deema AlBunyan.

**Software:** Mohammed Alabdulmuhsin, Deema AlBunyan.

**Supervision:** Monther Alsultan.

**Writing – original draft:** Monther Alsultan.

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
