## [Decision Letter · Decision Letter 0]

10 Dec 2024

Response to Reviewers
Revised Manuscript with Track Changes
Manuscript
**Journal Requirements:**

1. We ask that a manuscript source file is provided at Revision. Please upload your manuscript file as a .doc, .docx, .rtf or .tex.

2. Please provide separate figure files in .tif or .eps format.

For more information about figure files please see our guidelines:https://journals.plos.org/digitalhealth/s/figures 

**Additional Editor Comments (if provided):**
**Reviewers' Comments:**

**Comments to the Author**

1. Does this manuscript meet PLOS Digital Health’s publication criteria?

Reviewer #1: Yes

2. Has the statistical analysis been performed appropriately and rigorously?

Reviewer #1: N/A

3. Have the authors made all data underlying the findings in their manuscript fully available (please refer to the Data Availability Statement at the start of the manuscript PDF file)?

Reviewer #1: Yes

4. Is the manuscript presented in an intelligible fashion and written in standard English?

Reviewer #1: Yes

Reviewer #1: Your work presents an innovative approach to addressing medication safety and healthcare accessibility challenges through digital technology. Your implementation of an AI-powered platform for warfarin interaction checking demonstrates strong technical merit and clear clinical utility. The integration of multiple input modalities and bilingual support directly addresses important healthcare needs. The user interface design, particularly the color-coded interaction severity system, effectively makes complex medical information accessible to diverse users.

To strengthen the manuscript for publication, please address the following minor revisions:

Title and Formatting: Please review the manuscript title for grammatical consistency, particularly regarding hyphenation and capitalization. Consider revising to "Development of an Artificial Intelligence-Enhanced Warfarin Interaction Checker Platform."

Methods and Results: Include brief quantitative metrics from your beta testing phase to support the platform's effectiveness. A concise summary of user satisfaction scores or system accuracy measurements would be valuable. Please add a short description of the validation methodology used to verify the system's accuracy in identifying drug interactions.

Technical Implementation: Provide a brief overview of the security measures implemented for handling medical information. This addition will address an important consideration for clinical implementation. If your goal is to demonstrate technical feasibility of the approach only, then this would not be needed, but if so, include a comment as to have this may be achieved in a clinical implementation.

These revisions will enhance what is already a strong manuscript. The fundamental research and implementation are sound, and the potential impact on clinical practice is significant. Your work makes a valuable contribution to digital health literature, and I look forward to seeing the revised version.

**Do you want your identity to be public for this peer review?** For information about this choice, including consent withdrawal, please see our Privacy Policy

Reviewer #1: **Yes: ** Dayanjan S. Wijesinghe

**Figure resubmission:****Reproducibility:** To enhance the reproducibility of your results, we recommend that authors of applicable studies deposit laboratory protocols in protocols.io, where a protocol can be assigned its own identifier (DOI) such that it can be cited independently in the future. Additionally, PLOS ONE offers an option to publish peer-reviewed clinical study protocols. Read more information on sharing protocols at https://plos.org/protocols?utm_medium=editorial-email&utm_source=authorletters&utm_campaign=protocols

---

## [Decision Letter · Decision Letter 1]

17 Jan 2025

Development of an Artificial Intelligence-Enhanced Warfarin Interaction Checker Platform.

PDIG-D-24-00492R1

Dear Dr ALSULTAN,

We are pleased to inform you that your manuscript 'Development of an Artificial Intelligence-Enhanced Warfarin Interaction Checker Platform.' has been provisionally accepted for publication in PLOS Digital Health.

Best regards,

Jennifer N Avari Silva, MD

Section Editor

PLOS Digital Health

**Additional Editor Comments (if provided):**

**Reviewer Comments (if any, and for reference):**

Reviewer's Responses to Questions

**Comments to the Author**

Reviewer #1: All comments have been addressed

publication criteria?

Reviewer #1: Yes

3. Has the statistical analysis been performed appropriately and rigorously?

Reviewer #1: Yes

4. Have the authors made all data underlying the findings in their manuscript fully available (please refer to the Data Availability Statement at the start of the manuscript PDF file)?

Reviewer #1: Yes

5. Is the manuscript presented in an intelligible fashion and written in standard English?

Reviewer #1: Yes

Reviewer #1: Thank you for addressing all of the criticisms. I think this will be a great addition to the state of current knowledge

**Do you want your identity to be public for this peer review?** For information about this choice, including consent withdrawal, please see our Privacy Policy

Reviewer #1: **Yes: ** Dayanjan S. Wijesinghe
